# A Systematic Review of Open Source Clinical Software on GitHub for Improving Software Reuse in Smart Healthcare

**Zhengru Shen ***[ID] **and Marco Spruit**[ID]

Department of Information and Computing Sciences, Utrecht University, 3584 CC Utrecht, The Netherlands; m.r.spruit@uu.nl

\* Correspondence: z.shen@uu.nl; Tel.: +31-30-253-6454

**Abstract:** The plethora of open source clinical software offers great reuse opportunities for developers to build clinical tools at lower cost and at a faster pace. However, the lack of research on open source clinical software poses a challenge for software reuse in clinical software development. This paper aims to help clinical developers better understand open source clinical software by conducting a thorough investigation of open source clinical software hosted on GitHub. We first developed a data pipeline that automatically collected and preprocessed GitHub data. Then, a deep analysis with several methods, such as statistical analysis, hypothesis testing, and topic modeling, was conducted to reveal the overall status and various characteristics of open source clinical software. There were 14,971 clinical-related GitHub repositories created during the last 10 years, with an average annual growth rate of 55%. Among them, 12,919 are open source clinical software. Our analysis unveiled a number of interesting findings: Popular open source clinical software in terms of the number of stars, most productive countries that contribute to the community, important factors that make an open source clinical software popular, and 10 main groups of open source clinical software. The results can assist both researchers and practitioners, especially newcomers, in understanding open source clinical software.

**Keywords:** open source; GitHub; clinical software; systematic study; generalized additive models; topic modeling

## 1. Introduction

There have been numerous open source projects created across a variety of domains after decades of open source software advocacy [1]. With their accelerating popularity, more and more will likely be added. The plethora of open source projects offers great reuse opportunities for developers to build tools at lower cost and at a faster pace. Therefore, software reuse, also called code reuse, has become an essential topic in software development [2–4]. This is especially true in the clinical domain, where the lack of a well-trained IT workforce and an uneven geographic distribution of clinical informaticians between developed and developing countries restrain the development of clinical IT [5,6]. Given that successful reuse of existing open source software enables the rapid and cost-effective development of clinical applications, it is of great importance to promote reuse awareness in the clinical community and facilitate software reuse practices among clinical developers and applied data scientists [7–10]. Zhang and Ho have recognized this importance and have called for more reuse of open source projects in clinical settings [7].

As a challenging topic, software reuse still faces many difficulties, particularly in clinical software development [11]. The lack of resources to support the selection of suitable reusable source codes or

software from tens of thousands of open source projects is one of them. Although systematic literature studies on clinical software have helped developers obtain a deeper understanding of the existing tools and their performances [12–14], they are not particularly useful in clinical software development. First, systematic literature studies cover only a small proportion of the vast amount of clinical software. Moreover, the rapid updates of clinical software are not reflected due to the long publication process. Most importantly, these systematic literature studies do not examine source codes, software issues, and code usage, which are crucial in reuse practices.

Recently, researchers in life sciences have combined literature studies and large-scale analysis on open source repositories to achieve better reusability [15,16]. A source code repository is a file archive and web hosting facility where a large amount of source code is kept, either publicly or privately [17]. Well-organized repositories contain both brief descriptions and detailed README files that explain what the projects are and how to use them. Quantitative metrics such as stars and forks indicate how popular a repository is. Wang et al. built an online software discovery platform with biomedical software-related data collected from PubMed literature and GitHub. It empowers biomedical researchers to easily find the (open source) tools they need [15]. A large-scale analysis of bioinformatics open source projects on GitHub was conducted to uncover the unknown state of bioinformatics software. The analysis covers a list of 1743 GitHub repositories manually selected from bioinformatics articles and online forums [16]. By focusing on the analysis of data extracted from open source repositories, these studies offered more information for evaluating software in reusability. However, both studies only included a limited amount of biomedical software that were mentioned in the literature. To our knowledge, there is no such research that has investigated open source clinical software.

Therefore, this paper intends to contribute to clinical software reuse research by conducting a large-scale analysis of open source repositories in the clinical domain. Instead of analyzing a shortlist of open source repositories extracted from literature, this study includes all clinical-related open source repositories on GitHub. As the largest code host in the world, GitHub reached 24 million developers working across 67 million repositories in 2017 [18]. Numerous open source projects in various domains could be found and reused. A systematic study on GitHub repositories could provide a good representation of available open source tools in clinical domains. Besides, GitHub offers an easy application programming interface (API) for external users to retrieve data from repositories.

Natural language processing (NLP) quantitative research via topic modeling was conducted to uncover the distribution of open source repositories over different topics. Additionally, quantitative analysis of numerical data revealed the status of repositories in terms of their reuse capability. The analyses provide sufficient information to support clinical developers' decisions in the software development process. Moreover, this study developed a reproducible and scalable method for systematic studies on GitHub repositories. Reproducing the research can be completed by running the data collection and analysis scripts we host on GitHub. To scale up the study to other domains, one merely needs to customize the search terms.

In particular, the purpose of the paper is to shed light on the following questions, so that clinical developers can make more informed decisions:

- What is the current status of open source clinical software?
- What are the impacts of various factors, such as the number of contributors and the number of commits, on the popularity of an open source clinical software?
- What are the main focus areas of all the collected open source clinical software?

The paper is structured as follows. Section 2 illustrates a number of methods used in our study. Then the results are presented in Section 3. Discussions in Section 4 summarize the main findings of this research and list some limitations. Section 5 concludes the study.

## 2. Materials and Methods

This study underwent two main steps: (1) Data collection that extracted data from GitHub repositories via the GitHub REST API v3 [19], and (2) data analysis in which both numerical and textual data were analyzed to reveal insights on open source repositories in the clinical domain. The data collection produced the core material of this study, GitHub data. To uncover the current status, we applied descriptive analysis to obtain a summary of open source clinical software from many aspects. Second, since generalized additive models (GAMs) have an interpretability advantage, the study employed a GAM to explore the relationships between various factors and the number of stars of an open source clinical software. Lastly, the description of an open source clinical software offers a quick way of understanding what it does and what it is for. Therefore, the main focus areas were derived from thousands of descriptions with topic modeling.

### 2.1. Data Collection and Processing

### 2.1.1. Search Terms

This study refers to clinical software as software that is implemented in healthcare to help doctors improve patient care. Software that is developed for biological research, such as genome sequencing [20] and cell screening [21], were excluded. Therefore, "(clinical OR medical) OR (patient OR doctor)" was selected as the search term. English terms were used given that the majority of repositories have their names and descriptions in English. Figure 1 shows the numbers of repositories returned using the search term "clinical" in different languages. Second, the search term covered the main entities in the clinical domain.

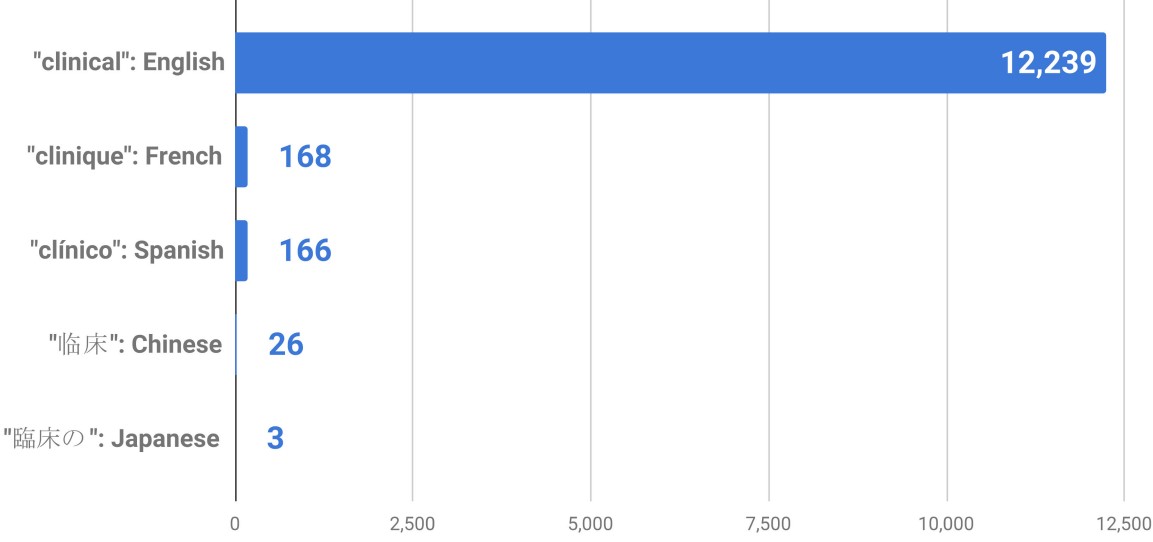

**Figure 1.** The number of repositories returned with the search term "clinical" in different languages.

### 2.1.2. Data Extraction

GitHub REST API v3 exposes GitHub repository data to external users. Its search API offers an optimized solution for users to locate the specific items that interest them most, such as repositories, users, and issues. GitHub repositories in this study were obtained by using the search repository with the above term, and data for each repository including name, description, README files, stars, forks, the number of contributors, and the number of commits were extracted. Table 1 displays the features extracted and the purposes of extracting them.

**Table 1.** Data extracted from GitHub REST API v3.

| Data Type | Data Extracted | Description |
|---|---|---|
| Textual | Description | A brief explanation of what a repository does, what it is for, and other items. |
| | README files | A README file summarizes the most important aspects of your project. It generally includes project name, description, table of contents, installation, usage, contributors, credits, and license. |
| | Languages | Programming languages of the source codes |
| | Owner type | Two types of owners exist: Organization and individual developers. |
| | Owner location | Location of an owner. |
| Numerical | Stargazers | Stargazers refers to the number of times a repository is bookmarked. It reflects an approximate level of interest in the repository. |
| | Forks | A fork is a copy of a repository. Forking is necessary for developers to contribute a project. Forks refers to the number of forks. |
| | Contributors | The number of contributors who have worked for a repository. |
| | Commits | The total number of commits. |
| | Created date | The date that a repository was created. |
| | Updated date | The latest date that a repository was updated. |
| | Issues | Issues is the number of open issues in a repository. |
| | Size of source codes | The size is valued as the size of the whole repository (including all of its history), in KB. |
| | Size of README file | The size of the README file of a repository, in B. |

### 2.1.3. Automated Data Pipeline

The data extraction pipeline was written in Python using a third-party library, PyGitHub. The pipeline took the chosen search terms as input and received repository data in JSON. The JSON responses were first filtered and then converted to database records and pushed to tables in a MySQL database. Repositories with no description or no programming language specified were excluded from further analysis for the reason that clinical software was the focus of our study. The whole process is reproducible by running the Python scripts at Reference [22]. Moreover, replacing the search term with others scales the pipeline to other domains.

### 2.1.4. Text Processing

Although GitHub has allowed users to attach topics as labels of the repositories since 2017, only a small fraction of them have topics. Specifically, only 6.6% (982/14,971) of the collected repositories have topic labels. Therefore, we utilized keywords extracted from a repository description as labels. Keywords extracted from the description presented an insightful indication into a repository. An external API, namely IBM Watson Natural Language Understanding [23], was requested to process the descriptions, which yielded a list of informative keywords for each repository.

Nevertheless, the extracted keywords could not be directly added as labels because a large number of keywords had semantically similar forms. Examples included "patients' data" versus "patients' information", "medical appointments" versus "doctor appointments", and "diabetes patients" versus "diabetic patients". To address this semantic issue, we employed a normalization process in which semantically similar keywords were combined. The process consisted of three steps: (1) Calculating the semantic similarity between keyword pairs, (2) labeling keyword pairs based on their similarity, and (3) replacing keywords with their semantically similar keywords.

To calculate the semantic similarity, this paper utilized Word2Vec [24] to represent keywords. Word2Vec is an unsupervised algorithm developed by Google that tries to learn meaningful vector representations of words from a dataset of text [24]. It does so based on the distributional hypothesis, which states that words that appear in the same context probably have a similar meaning. With Word2Vec, each keyword becomes a word vector that is compared to other keyword vectors to generate a similarity score. Based on their similarity, we divided keyword pairs into two groups, equal and tag. The threshold of the similarity score was 0.85. Keywords with a similarity score greater than or equal to 0.85 were considered as an equal keyword pair. Otherwise, they were a taggable pair. For equal keyword pairs, we replaced one of the keywords, the less frequent one, to replace the other. The "tag" meant that the more frequent keyword was added as a tag to the other. Table 2 shows some prominent examples of these keyword pairs.

**Table 2.** Examples of 10 keyword pairs based on their semantic similarity.

| Original Keyword | Replacement | Tag |
|---|---|---|
| Patients' information | Patients' data | Patients |
| Electronic medical record | Medical records | |
| Clinical data | Patients' data | Data |
| Medical information | Medical data | Medical records |
| Medications | Medicines | |
| Breast cancer patients | Cancer patients | |
| Healthcare | Medical care | Health |
| Pharmacists | Physicians | |
| Clinical studies | Clinical research | Clinical data |
| Simple web application | Web application | Application |

### 2.2. Descriptive Analysis

Descriptive analysis is applied to acquire the characteristics of a collection of data [25]. Together with simple graphics, it presents knowledge hidden in large datasets in a manageable way. To identify interesting characteristics that reflected the current status of the GitHub repositories, we conducted a descriptive analysis. Specifically, we identified popular repositories in terms of stars, most rapidly growing repositories according to stars per day, most diverse repositories in terms of the number of contributors, and most active repositories in terms of the number of commits per day.

Graphics are of great importance to descriptive analysis [25]. A graphic was created with geographic visualization to show the geographic distribution of open source clinical software. Geographic visualization refers to a set of techniques for analyzing spatial data [26]. The graphic reflected the current development of clinical software across countries, which provided some insights into the uneven geographic distribution of clinical informaticians between developed and developing countries.

### 2.3. Generalized Additive Models

As an extension of generalized linear models (GLMs), a GAM is an additive modeling technique that captures the impact of the predictive variables through smooth functions [27]. It is best known for its interpretability, which can determine the contribution of each independent variable to the prediction. Therefore, we chose a GAM to help us identify the relationships between several repository features and its popularity (the number of stars). To begin with, we built a linear regression model that predicted the number of stars of a repository based on a number of predictors: Forks, the number of issues, the repository size, owner type, the size of README files, the number of contributors, the total number of commits, and creation year. Then, partial dependency plots were produced to explicitly show the relationships between the predictors and the number of stars on GitHub repositories. We chose pyGAM, a Python package for GAM, as the package for our implementation [28].

### 2.4. Topic Modeling

For broad search queries, thousands of unique keywords may remain, even after the previously described similarity-based processing step. Therefore, labeling the repositories with such keywords could not provide a good overview. Our aim was to identify a few numbers of broader topics that could be used to label repositories. This paper applied an unsupervised machine learning approach, latent Dirichlet allocation (LDA), to infer topics from the extracted keywords. We set 10 to 20 as the range in creating optimal LDA models.

LDA, first introduced by Blei, Ng, and Jordan in 2003, is a generative probabilistic topic model that aims to discover the underlying topics from a corpus [29,30]. LDA models document probabilistic distributions over *K* latent topics, where each latent topic is represented by a probabilistic distribution over words from a fixed vocabulary. The words with the highest probabilities in each topic usually give a good indication of what the topic is. As a robust open source topic modeling toolkit, Gensim offers an easy way of implementing topic modeling [31]. We implemented our LDA with Gensim in Python [32].

We first trained an LDA model on the extracted keywords from all the repositories, and then five more LDA models were built on repositories from different periods: Prior to 2015, 2015–2016, 2016–2017, 2017–2018, and post-2018. Each of these periods contained a roughly similar number of repositories, around 3000. Table 3 lists all the models and the parameters that generated the optimal result in terms of the cohesion and explainability of topics. Alpha dictates how many topics a document potentially has. The lower the alpha, the lower the number of topics per document is. Passes is the number of times you want to go through the entire corpus.

**Table 3.** Latent Dirichlet allocation (LDA) models: Five models that built on yearly repository data.

| LDA Model | Number of Repositories (Repos) | Number of Topics | Alpha | Passes |
|---|---|---|---|---|
| All repositories | 14,971 | 10 | 0.001 | 10 |
| Prior to 2015 | 2784 | 8 | 0.002 | 6 |
| 2015–2016 | 1899 | 6 | 0.01 | 8 |
| 2016–2017 | 2872 | 8 | 0.01 | 6 |
| 2017–2018 | 4330 | 8 | 0.02 | 8 |
| Post-2018 | 3086 | 8 | 0.01 | 10 |

### 2.5. Interactive Data Visualization

Data visualization empowers researchers to present their results in a graphical manner [33]. In this paper, we visualized our findings with interactive data visualizations, which are accessible online. Interactive data visualization engages more with users by allowing them to select specific data points to visualize the data in the way they choose [34]. In particular, radar charts and time-series line plots were created with Plotly.py [35], an interactive, browser-based graphing library for Python. In addition, we built an interactive top modeling visualization with pyLDAvis to improve the interpretation of the generated topics [36,37].

## 3. Results

This section presents the main findings of our analysis, which answers the questions described in Section 1.

### 3.1. Current Status

The overview provides a quantitative summary of all collected GitHub repositories. It highlights some important characteristics of the dataset and illustrates the current status of open source clinical software from different aspects. First of all, there have been 14,971 clinical-related GitHub repositories created since 2008. Among them, 12,919 are open source clinical software for the reason that they own publically accessible source codes. Moreover, there has been an upward trend in the number of

repositories created on a yearly basis. Until 2011, the number of repositories was around 100 per year. The number rapidly increased to around 1000 from 2012 to 2014. Since the average yearly growth rate over the last five years has been 55%, the number of repositories created in a year will reach around 9300 in 2020.

Table 4 lists repositories according to a variety of metrics. From "popular repos", "most rapidly growing repos", "10 frequency topics extracted from repos", and "top 10 libraries", we found that clinical software on machine learning, medical images, and electronic medical records attract more attention from developers. The most rapidly growing repositories list indicated that deep learning-related repositories are becoming dominant, with five out of seven of the most rapidly growing ones addressing deep learning issues in the clinical field. Nevertheless, the most diverse repositories and most active repositories did not show a clear pattern.

**Table 4.** Overview of open source clinical software.

| Characteristics | Statistics |
| --- | --- |
| # of repos that were not empty | 14,971 |
| Open source clinical software (# of repos with codes) | 12,919 |
| Top 10 countries (# of repos) | U.S., India, United Kingdom, Canada, China, Germany, France, Bangladesh, Australia, Brazil |
| Popular repos (# of stars) | ResearchKit, openemr, ClearCanvas, dwv, cornerstone, Deep-Learning-for-Medical-Applications, CTK, NiftyNet, 3DUnetCNN, OpenClinica |
| Most rapidly growing repos (# of stars per day) | ResearchKit, NiftyNet, DLTK, Attention-Gated-Networks, Deep-Learning-for-Medical-Applications, chart-doctor, 3DUnetCNN |
| Most diverse repos (# of contributors) | ResearchKit, openmrs-module-bahmniapps, medical-appointment-scheduling, sweetdoc, SwasthIndia, CTK, sofa-framework, MITK, dicom_tools |
| Most active repos (# of commits per day) | MITKats, openeyes, cds-stack, sofa-framework, myclinic-spring, renalware-core, BluePearlViewer, SwasthIndia, clinical-meteor-tool |
| Top 10 frequency topics extracted from repos | machine learning, healthcare, medical image, deep learning, clinical trial, patients, medical information, natural language processing |
| Top 10 libraries | React, Jquery, Angular, Tensorflow, Ajax, Redux, Keras, OpenCV, Pytorch, Scikit-learn |

### 3.2. Geographic Distribution

Table 3 shows the top 10 countries that contribute most to the open source community based on the number of GitHub repositories. Figure 2 is the geographic visualization of a country's repositories (accessible at Reference [38]). As shown in the figure, the U.S. is unsurprisingly the biggest player, taking up to 38.8% of the total GitHub repositories. Among developing countries, India contributes the most, with around 10%.

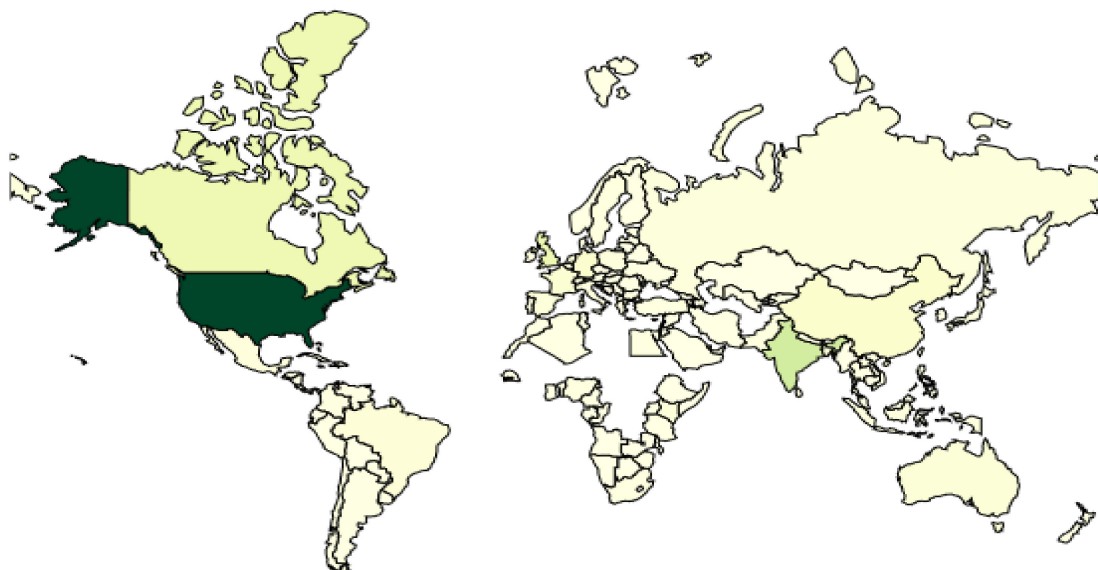

**Figure 2.** Geographic visualization of GitHub repositories by country.

### 3.3. Factors Impacting the Popularity of Open Source Clinical Software

Many factors affect the popularity of GitHub repositories. However, given the data we could collect, we mainly examined the impact of eight factors: The number of forks, the number of open issues, repository size, owner type, README file size, total contributors, total commits, and year of creation. The partial dependence plots, as shown in Figure 3, reveal the impacts of different factors on the number of stars. First, forks and the number of contributors exhibited a clear positive effect on the number of stars. A higher number of forks or contributors led to more stars. Meanwhile, the README file size had litter influence. Whether a repository is owned by an organization or individual developers was also trivial. With regard to the year of creation, we noticed that repositories that had existed significantly longer tended to earn more stars, but the impact diminished after 2010.

When the number of open issues was between 60 and 120, the number of stars was at a lower point. When the value was larger than 120, there was a positive correlation. A possible explanation is that the number of issues rises as developers of a repository stop further development, which then drives potential users away. Lastly, the impacts of repository size and the number of commits were unclear.

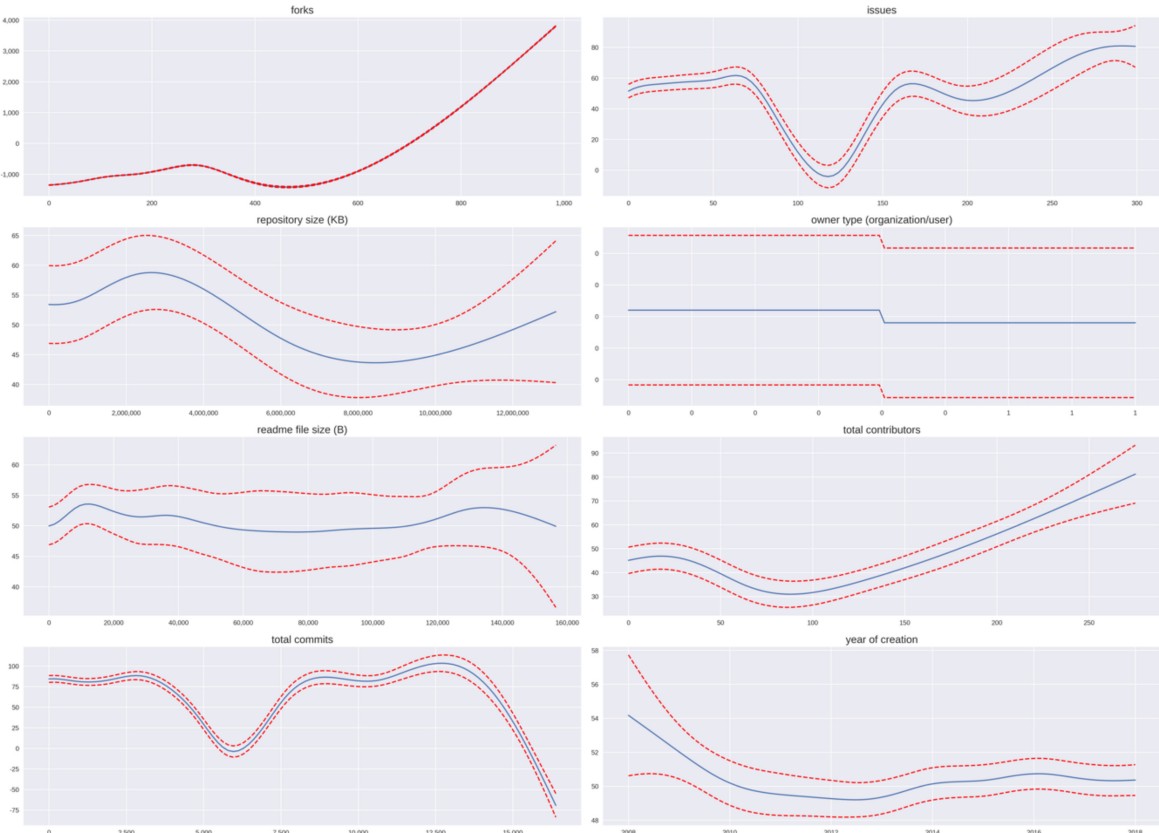

**Figure 3.** Partial dependency plots showing the relationships between the predictors (forks, open issues, repository size, owner type, README file size, total contributors, total commits, year of creation) and the number of stars on GitHub repositories.

### 3.4. Main Focus Areas

As shown in Figure 4, ten topics discovered in all repositories were as follows: Medical images analysis, applications for medical practice, medical data reuse, clinical text processing, applications for doctors, medical research, medical education, healthcare management systems, clinical trials, and medical data analysis. Table 5 offers a brief description of each of these topics and their top keywords.

Analysis of topics generated from the yearly data revealed that new technologies emerged over time. For instance, the open source community has devoted a lot to medical images analysis since the very beginning. It has also been a topic throughout all the years. Top keywords in the topic showed that deep learning and neural networks emerged as popular techniques in 2015. Moreover, we noticed that recently developers have been starting to evaluate blockchain in medical record sharing.

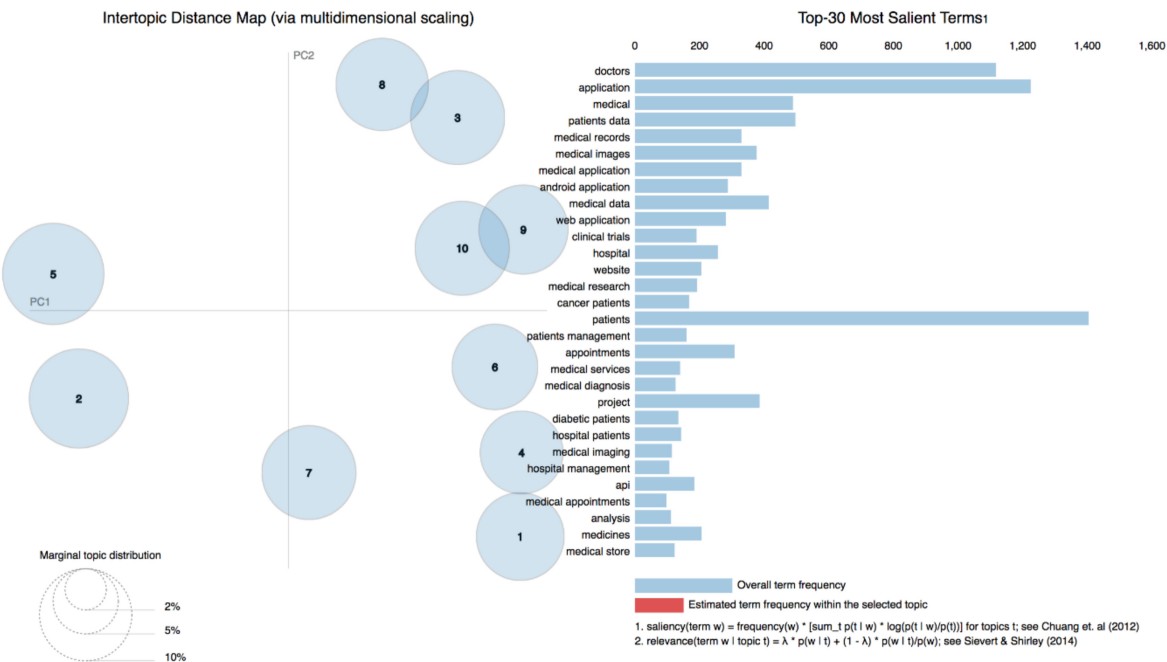

**Figure 4.** Interactive visualization of clinical topic clusters.

**Table 5.** Ten topics of all clinical GitHub repositories. API: Application programming interface.

| # in Figure 4 | Topic Label | Top Keywords |
| --- | --- | --- |
| 1 | Medical images analysis | Medical images, medical image processing, deep learning, medical image analysis, medical image segmentation |
| 2 | Applications for medical practice | Android application, medical applications, prescriptions, medical practice, appointments, medical care |
| 3 | Medical data reuse | Medical data, website, API, medical services, database, MySQL |
| 4 | Clinical text processing | Medical documents, symptoms, medical history, clinical notes, natural language processing (NLP) |
| 5 | Application for doctors | Doctors, applications, doctor appointments, medical professionals, doctor office, nearby doctor, medical staff |
| 6 | Medical research | Medical research, clinical research, clinical data, patient registration, projects, design, clinical decision support, medical informatics |
| 7 | Medical education | Medical students, final project, medical center, project Gutenberg book, experiments, paperwork |
| 8 | Healthcare management systems | Patient management, medical appointments, medical store management, patient monitoring, patient management systems, platform, software |
| 9 | Clinical trials | Clinical trials, patients, data analysis, new patients, survival analysis |
| 10 | Medical data analysis | Patient data, medical imaging, machine learning, model, data science, gene expression, predictive model |

## 4. Discussion

With the purpose of shedding light onto the vast amount of clinical-related open source software, our study collected data through the GitHub API and then conducted various analyses to uncover both overall and specific characteristics among GitHub repositories. The analysis methods included a descriptive analysis, trends analysis, and topic clustering. The main findings are summarized as follows:

- The number of repositories and the growth rate indicated an increasing interest, which suggests the potential of utilizing open source software in both academic and practical clinical settings;
- Clinical software on machine learning, medical images, and electronic medical records attract more attention from developers, and deep learning-related repositories are becoming dominant;
- The U.S. is the biggest player in the world, taking up to 38.8%. Among developing countries, India contributes the most, with around 10%;
- Factors that affect the popularity of a GitHub repository include forks and the number of contributors. The README file size and owner type had litter influence. Other factors, such as the number of open issues, the repository size, and the number of commits, are unclear.
- The LDA model clustered keywords extracted from repository descriptions into 10 groups, ranging from clinical text processing to medical education. Moreover, analysis of the topics generated by yearly data revealed that topics evolve over time.

Besides the above findings, our study contributes to the scientific community by providing a reproducible method of studying GitHub repositories. Over the years, researchers have been struggling with reproducing research results such as literature studies. As mentioned above, both data collection and analysis could be reproduced with a rerun of the Python scripts. Scaling up the approach to other domains requires some additional inputs, such as search terms, and the number of topics for topic modeling.

Nevertheless, some limitations to this study should be noted. Although GitHub is a major platform where developers work and share their codes, there are other platforms that are also of great importance to the open source community, such as SourceForge and GitLab. Collecting more data from such platforms may therefore produce more complete and deeper insights.

Furthermore, README files were also employed to extract keywords because some repositories only describe coarsely what the code does in the project description. Keywords extracted from README files could provide more information so that the following topic modeling would be more accurate. For example, borochris/CLinUiP has a very short and uninformative description, "Clinical UI Portal", but its README file contains a clear explanation, "Clinical UI Portal is an Open Source demonstration medical portal".

Last but not least, the topic labeling was limited. First of all, some of the topics contained overlapping keywords, given that they were related. For instance, medical image analysis has "medical images" as a top keyword, and "medical imaging" belongs to medical data analysis. In addition, the labels were manually assigned by authors only, instead of a group of experts.

## 5. Conclusions

This paper provided an overview of the open source GitHub projects in clinical domains with the purpose of understanding the status of open source clinical software. It, on the one hand, helps clinical developers be aware of the status quo so that they can make an informed decision while they design their systems and tools. Instead of creating all code from scratch by investing a lot of resources and time, making good use of existing tools and code can be a more efficient strategy. On the other hand, researchers might get possible research direction from this study.

**Author Contributions:** Z.S. conducted the research, including the research design, data collection, and data analysis. M.S. supervised the research and reviewed the paper. All authors read and approved the final manuscript.

**Funding:** This work is part of the project "OPERAM: OPtimising thERapy to prevent Avoidable hospital admissions in the Multimorbid elderly", supported by the European Commission (EC) HORIZON 2020, proposal 634238, and by the Swiss State Secretariat for Education, Research and Innovation (SERI) under contract number 15.0137. The opinions expressed and arguments employed herein are those of the authors and do not necessarily reflect the official views of the EC and the Swiss government.

**Conflicts of Interest:** The authors declare no conflicts of interest.

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
