# Peer review of "A Systematic Review of Open Source Clinical Software on GitHub for Improving Software Reuse in Smart Healthcare"

_applsci, doi:10.3390/app9010150_

Reviewer 1 Report

This manuscript describes the clinical open source trends in GitHub. Interesting topics, but need more analysis. 

Major comments: 

1) Table 3 shows diverse characteristics of open source clinical SW. However, lots of characteristics are useless to researchers. As a researcher, I cannot have any meaningful information using "Top 10 programming languages," "individual vs. organization", "Top 10 libraries", "Top 10 contributors in terms of # repo.". These features do not guide any research direction, they are only valuable for some news articles. 

In addition, the number of forks and the number of stars will give more information to readers. 

2) The authors only give a descriptive analysis using statistics and LDA. Need more in-depth analysis for the chosen repository. What are the precise target of repo?, what kind of data they use? How the users can prepare the data set?

3) What is the meaning of Hypothesis in page 5? Why do the authors want to know the size of readme file and the count of stargazers?

4) It is not clear GitHub repo only deals with clinical data set are included or not. Please explain it. 

5) Figure 2 is meaningless. JS is the dominant PL. So what? 

Similarly, Figure 3 gives no interest result as well. 

6) Figure 4, there is no red-bar in the legend. 

And what is the meaning of left-bottom figure? 

Minor comments

7) The description of LDA in pages 5 ~ 6 should be shortened. Giving the reference is enough. 

8) There are typos. i.e., (In Abstract), First, we first develop

Author Response

uploaded as a WORD file

Reviewer 2 Report

The paper explores development tools for software in medicine. Especially the open source approach has great potential for further innovation. The paper therefore may contribute substantially to progress in the field. the authors may however add substantial value to this work, by addressing some issues in this version of the paper. This may increase the benefit for the readers substantially.

General comments:

The described methods do not clearly relate to the field of interest of the work. The authors may improve the benefit to readers by providing clear arguments, why each method contributes to answering the questions of the work. Also there are some issues on the methods, that better be addressed, see specific comments.

Specific comments:

Abstract:

"The results include the most 12 successful open source clinical software, the trend, and the classification in terms of their clinical 13 purposes." The terms "successful", "trend" and "classification" demand a clearer definition. It might help to review and revise the paper to assure that they are clearly defined and used throughout the paper.

"First, we first develop ...": Typo, delete one "first"

Introduction

"Given that successful reuse of existing open source projects ...": The object of reuse is probably not the "project", but "the sourcecode developed within a project". Please consider to revise.

"The lack of resources to support the selection of suitable reusable codes": The term "codes" here may relate to code systems / vocabularies, or to software source code. Does it make sense to use the term "sourcecode" consistently throughout the paper? Please consider.

"API", "NLP quantitative research via topic modeling" Please spell out acronym at first use.

"Reproducing the research can be completed by simply running the data collection": "simply": How do you define simple? Consider to remove. Review the complete paper accordingly.

"In particular, the purpose of the paper intends to shed light": replace "intends" by "is"?

"In particular, the purpose of the paper intends to shed light on the following questions": The four listed questions are in part not specific enough to form the base for this paper. Readers will find it hard to understand how the authors derive the specific methods and results used in this work from these questions. Please revise to provide a clear relation between "questions" and methods.

Materials and Methods

2.1. Data collection
2.1.1. Search terms

"This study refers to clinical software to software that implemented in clinical practices to helps doctors improve patient care." Typos, please revise.

Same sentence: "implemented in clinical practices": Consider to replace "clinical practices" by "health care"?

"Biomedical software, such as genome sequencing ...": Many readers will assume that "Biomedical software" includes bioinformatics software. Please revise, to assure consistent use of terms.

Data extraction

"GitHub REST API v3 exposes GitHub ..." please provide reference

Table 1: There are other valuable parameters to describe activity on Git Hub: Number of users, number of commits, commits per user, number of most active users, downloads, geographical distribution of users. Why were these not considered? 

table 1: "The date where a repository is created.", "The latest date where a repository is updated.": change to "... when a repository was created/ updated"?

Also table 1: "Issues is the number of open issues in a repository. It indicates its activeness, with higher number of issues meaning higher number of active users." How do you argue that the number of issues indicates activity? If the code in a repository is not further developed, the number of issues may rise, without any activity from developers?

Automated data pipeline

"The JSON responses were first filtered and ..." Which criteria were used for filtering?

"Source code of the data pipeline is openly accessible at ... " Pleae provide reference, move URL to list of references.

"To address this semantic issue, we employed a normalization process in which semantically similar keywords are combined." How was this process validated? How do you define "semantically similar"?

"IBM Watson" "Word2Vec" Please provide refereces.

Table 2:

""Breast cancer patients" - "Cancer patients": Breast cancer is one type of cancer. May it be better suited to use "Cancer patients" as keyword?

"Pharmacists" - "Physicians": Do the authors indicate that these two professions are "equal"? If yes, this may be a severe weakness in the method. Please revise, and re-consider the method of "semantic similarity". See above, how did the authors validate this method?

Statistical analysis

Line 144-161: The authors provide a very baasic introduction into statistical analysis. This is not useful here. It is sufficient to provide reference to the methods used, as they are well-established.

Also review for formatting errors, e.g. line spacing.

"Spearman rank correlation": Was this the only statistical method used? On which data was it used? 

Interactive data visualization

"to visualize the story": Instead of "story"  - use "results" or "data"?

"Plotly.py": provide reference

Overview of the open source clinical software

"individual developers own 203 88.1% of the repositories," versus 88% in table 3: Assure that the numbers in the text exactly match those in the table.

Trend analysis

"Since the average yearly growth rate over the last 209 five years is 55%, the number of repositories is expected to reach around 9300 in 2020. " The study collected 14971 repositories (see line 118). the number of 9300 therefore is hard to understand here. This is a contradiction. Please resolve.

Figure 2: Which parameter does the y axis show? Please add axis labels and units.

Statistical analysis

"... the testing p-value was 234 3.1369275247169935e-175... " Why list all the many digits?

Topic modeling

This section describes parts of the method used. The authors might add value by moving this section to "methods".

Table 5:

- the "Medical images analysis" topic label includes "Medical images"

- the "Medical data analysis" topic includes "medical imaging"

These reults indicate that the automated clustering of keywords may need improvement. Please comment on this, at least in the discussion.

Conclusions

"It, on the one hand, helps bioinformaticians be aware of .." The paper explicitly excludes "Biomedical software, such as genome sequencing and cell screening". Why are bioinformaticians explicitely mentioned here?

Author Response

upload as a WORD file

Round  2

Reviewer 1 Report

The authors successfully addressed the issues I raised. They substantially revised the manuscript contents. At this stage, this manuscript can be published. 

Reviewer 2 Report

The revision has addressed the main issues adequately.

For the future it suggests itself to revisit this space frequently, to follow the very intensive effort in the field of software development. Also, additional views, for example on software quality and regulatory aspects may be of interest. The field will keep changing substantially. Structured analyses of the progress during this evolution over time may generate value.